# Spatial Effects of Urban-Rural Ditch Connectivity Gradient Changes on Water Quality to Support Ditch Optimization and Management

**Chunqi Qiu** [1], **Yufeng Li** [1,*], **Alan L. Wright** [2], **Cheng Wang** [3], **Jiayi Xu** [1], **Shiwei Zhou** [1], **Wanchun Huang** [1], **Yanhui Wu** [1], **Yinglei Zhang** [1] and **Hongyu Liu** [1]

1   School of Marine Science and Engineering, Nanjing Normal University, Nanjing 210023, China; 17805804382@163.com (C.Q.); vitas854878919@163.com (J.X.); svchouw9894@163.com (S.Z.); huangwc98@163.com (W.H.); wuyanhui_0105@163.com (Y.W.); zhangyinglei668@163.com (Y.Z.); liuhongyu@njnu.edu.cn (H.L.)
2   Soil & Water Sciences Department, University of Florida-IFAS, Gainesville, FL 34945, USA; alwr@ufl.edu
3   School of Geography, Nanjing Normal University, Nanjing 210023, China; wangc0602@163.com
*   Correspondence: pandalee_0826@163.com

**Abstract:** Ditch networks play crucial roles in regulating water fluxes with their surroundings. The connectivity of ditches can have great impacts on nutrient migration and transformations. However, connectivity patterns related to ditch networks have rarely been studied, especially the relationships with water quality assessed through spatial analysis. This paper considered ditch connectivity and water quality indicators comprehensively, using spatial autocorrelation and geographically weighted regression (GWR) models, to analyze the impact of ditch connectivity on water quality from urban to rural gradients. The results suggested that water quality in rural areas and towns was better than in suburbs and transition zones, and the different areas exhibited variable spatial ditch connectivity. The Moran's I index of the connectivity indicators showed the clustering state of spatial distribution, with ditch connectivity explaining 61.06% of changes in water quality. The circularity and network connectivity of the ditches had the most influence on water quality. However, the degree of influence varied with region. Circularity had the greatest impact on water quality in urban areas, and network connectivity had the greatest impact on water quality in township areas. Therefore, future water improvement projects, based on ditch optimization and management, need to consider the more related influencing factors and their spatial differences.

**Keywords:** urban-rural gradient; ditch connectivity; spatial autocorrelation; GWR model; water quality





## 1. Introduction

Agriculture developed thousands of years ago in plains areas of China, while the ditch area in the agricultural landscape accounts for about 10% of the acreage. As a basic part of the agricultural infrastructure, ditches play an important role to retain high agricultural water outputs through timely flood drainage and drought relief. In the precipitation-runoff process, water flows gradually converge to the target water body through the ditches. During the water demand period of agriculture, the water resources in the ditches, which were derived from precipitation and surface runoff, can be used as the irrigation water source and returned to the farmland. Therefore, ditches maintain the health of the rural water environment and the balance of the ecosystem through regulation of water supply [1].

For a long time, research focused on water adjustment of the ditches in the plains irrigation area [2] and water environment purification functions of ditches [3,4]. The pollutants contained in the water flowing through ditches can be purified through sediment adsorption, plant absorption, microbial absorption, and degradation [5,6]. The ditch

sediment not only provides growth carriers and nutrients for microorganisms and aquatic plants, but also has a purification effect on nitrogen and phosphorus in the water bodies [7]. However, the performance of these ecological functions is closely related to the connectivity of plain ditches [8]. Flow direction and hydrological connectivity in the ditch are complex and changeable because of flat terrain and the management of water conservancy projects in plain area.

Major research about the influence of ditch connectivity on water quality is mainly divided into two components. One part is to explore the relationship among land use, ditch, and the target water body on the watershed scale, such as: Taihu Lake Basin, Chaohu Lake Basin, coastal, and other regional scale studies [9–11]. These studies emphasized the influence of ditches as a land use type on the water quality, and seldom discussed the effect of ditch connectivity characteristics on water quality alone. Second, with the development of cities and the deterioration of urban water quality in ditches, most studies focus on the relationship between the ditch connectivity and water quality in urban areas, and there are few studies focused on rural areas [12]. However, with the emergence of water quality problems in rural areas in China, research on the water quality of ditch networks has gradually expanded from urban areas to rural areas. Therefore, this study chose the northern Jiangsu plain irrigation area to analyze the differences in the characteristics of ditch connectivity under the urban-rural gradient, and to explore the spatial relationship between ditch connectivity and water quality through spatial autocorrelation analysis and geographically-weighted regression models. The study results can provide theoretical support for optimizing the water system structure in the plain water network area while optimizing scientific and reasonable water conservancy project scheduling and operation rules.

## 2. Materials and Methods

### 2.1. Study Area

The study area is located in Funing County, the middle of the Jianghuai Plain, Jiangsu Province, between latitude 33°40′~33°56′ and longitude 119°43′~119°50′. The transitional climate from northern subtropical to warm temperate zone is the main climate condition in this area. The plain is 16.8 km long from north to south, 9.4 km wide from east to west, and covers an area of 88.5 km². The average annual rainfall and temperature is 979.6 mm and 14 °C (0–27 °C), and the average rainy days are 106, with most rainfall concentrated in summer. The water conservancy facilities on the ditches include pumping stations, culverts, and sluices, with ditches accounting for 14.73% of the area. According to field statistics, there are 225 pumping stations, 333 culverts, and 38 sluice points in the study area. Because the closing of sluices will cut the connectivity of ditches, the area with cutting sluices forms a relatively closed ditch network system. Therefore, the study area was divided into 46 spatial units, according to the location of the sluices (Figure 1).

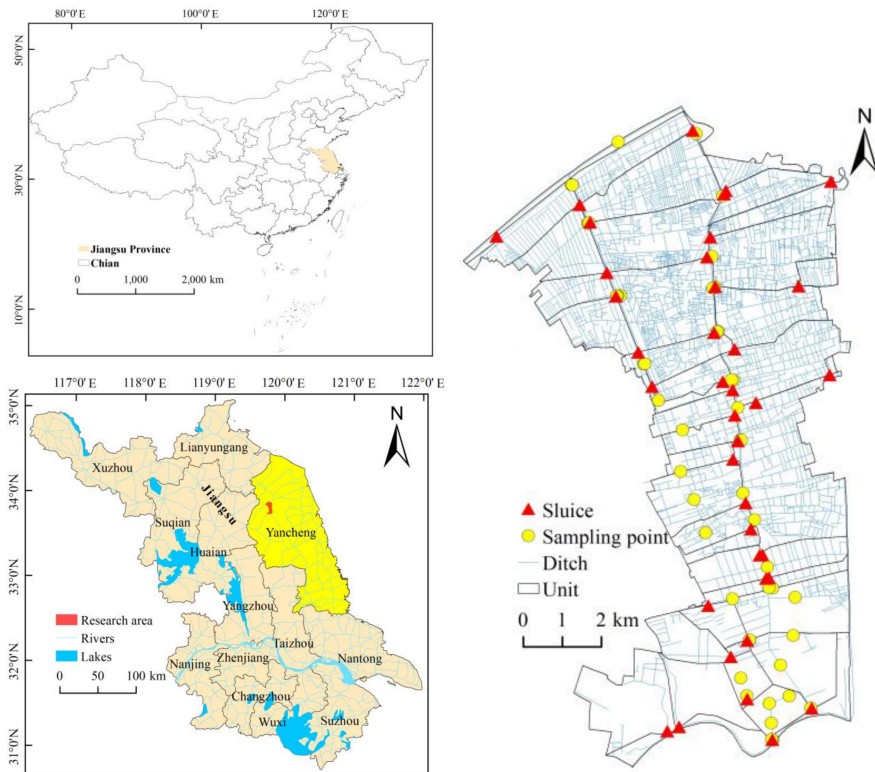

**Figure 1.** The location of the study area and distribution of sampling points.

### 2.2. Datasets

#### 2.2.1. Remote Sensing Data Processing

This research mainly focused on ditch connectivity characteristics under the conditions of urban and rural gradient changes. The width of the ditches was generally between 2–40 m. In order to identify the ditches and their water conservancy facilities more clearly, GF-2 remote sensing images with high resolution (1 m) were selected to reflect surface information of land patterns. Radiometric correction, geometric correction, and fusion of the images were preprocessed using ENVI5.2 software. Manual interpretation and cooperation, with supervised classification, were used to extract the vector data of land uses, ditch networks, and water conservancy projects in the study area. Although the study area was a plain, there were also micro-topography changes. Therefore, this study also used a 5 m resolution DEM (from the Department of Natural Resources of Jiangsu Province) to describe the micro-topography of the area.

#### 2.2.2. Urban-Rural Gradient Division

The ecological structure and processes exhibited significant differences between urban and rural areas. According to the characteristics of land use type and the degree of human utilization, this study constructed 5 regions extending from the urban to rural areas: urban area, suburb, transition zone, town, and rural area. Construction land was the main land use in the urban area, accounting for 67.16%, followed by paddy field, accounting for 13.13%. Paddy fields accounted for 47.50% in the suburbs, followed by construction land, accounting for 33.17%. The main land use type in the transition zones was paddy field, accounting for 84.81%. Paddy fields in the town accounted for 59.95%, followed by ponds and construction land, which accounted for 15.58% and 10.25%, respectively. In rural areas, paddy fields accounted for 63.03% of the land, followed by ditches and ponds, which accounted for 9.66% and 9.47%, respectively (Figure 2).

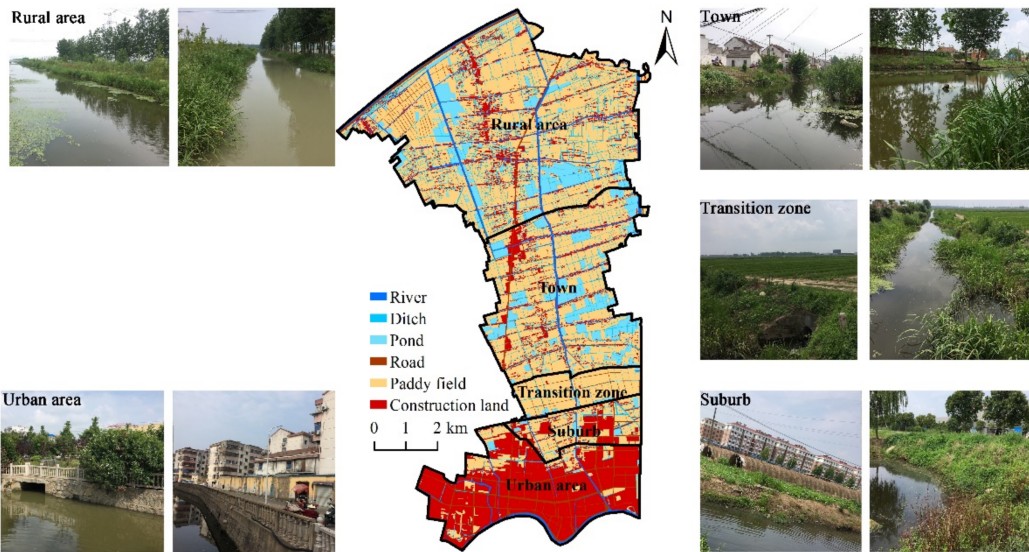

**Figure 2.** Land use and urban-rural gradient division map in the study area.

2.2.3. Water Quality Data Collection

(1)    Sampling sites.

In order to cover the entire study area and reflect the water quality characteristics, 47 sampling points were selected from urban to rural areas. The flow velocity of the ditches was minimal and every sampling point had 3 parallel samples. The summer rice and winter wheat rotation was the main land use type of agriculture, so the timing of field data collection was winter 2018 and summer 2019. Different numbers of sampling points were set according to the area of each sub-regime: 11 in the urban area, 3 in the suburb, 3 in the transition zone, 12 in the town, and 18 in the rural area.

(2)    Sample collection and lab analysis.

Surface water (0–20 cm) of 47 sample points was collected in 500 mL polyethylene sampling bottles and stored at 4 °C. The samples were taken to the laboratory for water quality assessment. The National Standard of the Republic of Groundwater Quality Standard (GB/T 14848-93) was used as the experimental and analysis standard.

A portable water quality analyzer (YSI 6600 of American Golden Spring Instruments, Beijing China) was used in the field to measure dissolved oxygen (DO) and pH. All water samples were analyzed for total phosphorus (TP), total nitrogen (TN), ammonia nitrogen (NH4+-N) and permanganate index (CODMn). The TP, TN, NH4+-N, and CODMn were determined in the laboratory according the Monitoring and Analysis Methods for Water and Wastewater [13]. The location (longitude and latitude) of each sampling point was georeferenced with a LT60 handheld GPS device (Huace Navigation Technology Ltd. China, location error approximately ± 0.8 m)

*2.3. Methodology*

2.3.1. Selection of Ditch Connectivity Index

The ditch network system was comprised of ditch corridors and connecting nodes from the perspective of landscape ecology. The ditch network structural analysis included three parts: the size of the ditch system, the complexity of the ditch network, and the influence of water conservancy projects on ditch connectivity. The length of the ditch (L), the ditch density (Id), and the number of corridors (LN) were selected for describing the size of the ditch system [14,15]. Circularity ($\alpha$), the ratio of corridors number to nodes number ($\beta$), and network connectivity ($\gamma$) were used to identify the complexity of the

ditch network [16]. The water conservancy project index ($\delta$) reflected the impact of water conservancy projects on the ditches in the study area.

(1) Ditch density ($I_d$): $I_d$ represented the length of the ditch per square kilometer. The ditch density was the basic index of ditch network structure. The greater the density of the ditch, the more frequent the interaction between land and water. The calculation formula is [17]:

$$I_d = \frac{\sum_{j=1}^{N} L_j}{A} \tag{1}$$

where $L_j$ is the length of the $j$-th ditch in the study area, $j = 1, 2, \ldots, N$; $A$ is the area of the study area.

(2) Circularity ($\alpha$): It can describe the choice of energy flow, logistics, and species migration routes in the network. It was a good indicator of network complexity and reflects the nutrient exchange capacity of nodes in the ditch network. The value of $\alpha$ is 0–1. When $\alpha = 0$, it means no loop; when $\alpha = 1$, it has the largest number of loops and the strongest nutrient exchange capacity. The calculation formula is [18]

$$\alpha = \frac{M - V + 1}{2V - 5} \tag{2}$$

where $M$ is the number of ditch corridors; $V$ is the number of nodes in the study area.

(3) Network connectivity ($\gamma$): It respected the degree of nodes connectivity in the network system. The range of $\gamma$ is 0–1. When $\gamma = 0$, it indicates that the nodes are not connected; When $\gamma = 1$, it means that each node is connected to other nodes, and the network connection is the best. The calculation formula is [19]

$$\gamma = \frac{M}{M_{max} \frac{M}{3(V-2)}} \tag{3}$$

where $M$ represents the number of ditches; $V$ is the number of nodes in the study area.

(4) Water conservancy project index ($\delta$): The pumping station was mainly used to pump water resources from lower areas to higher areas of the study area. Culverts in the ditches were mainly located below the road and when the width of ditches became narrow so the ditch connectivity was affected by culverts. According to the location of the water conservancy project and the levels of connected ditches, the weights are assigned by the expert grading method, with a large impact value of 0.7 and a less impact value of 0.3 [20,21].The calculation formula is:

$$\delta = 0.7a_1 + 0.3a_2 \tag{4}$$

where $a_1$ and $a_2$ represent the number of water conservancy projects with larger and smaller impacts in each calculation unit respectively.

### 2.3.2. Water Quality Index

Single factor index was the basis for developing the comprehensive water quality index. Different levels of water quality index corresponded to different ditch functions [22,23]. Water quality index was calculated to determine the ditch pollution and we selected DO, pH, $COD_{Mn}$, $NH_4^+$-N, TN, and TP to calculate the comprehensive water quality index. The calculation is as follows:

$$WQI = \sum_{i=1}^{n} \frac{WQI_i}{n} \times 100 \tag{5}$$

where $WQI$ is the water quality index, $n$ is the number of factors, $WQI_i$ is the $i$-th factor index. In addition, DO and pH have their own formulas below:

$$WQI_{DO,j} = \frac{DO_s}{DO_j}, DO_j \leq DO_f \tag{6}$$

$$WQI_{DO,j} = \frac{\left| DO_f - DO_j \right|}{DO_f - DO_s}, DO_j > DO_f \qquad (7)$$

$$DO_f = 468/(31.6 + T) \qquad (8)$$

where $WQI_{DO,j}$ is the dissolved oxygen index, $DO_j$ is the concentration of dissolved oxygen at the sampling point $j$ (mg·L$^{-1}$), $DO_s$ is the water quality standard of $DO$ (mg·L$^{-1}$), $DO_f$ is saturated point of $DO$ concentration (mg·L$^{-1}$), $T$ is the water temperature (°C).

$$WQI_{PH,j} = \frac{7.0 - PH_j}{7.0 - PH_{sd}}, PH_j \leq 7.0 \qquad (9)$$

$$WQI_{PH,j} = \frac{PH_j - 7.0}{PH_{sd} - 7.0}, PH_j > 7.0 \qquad (10)$$

where $WQI_{pH,j}$ is the index of pH, $PH_j$ is the measured value of pH at sampling point $j$; $PH_{sd}$ and $PH_{su}$ are the low limit and high limit of pH in the standard. The water quality was classified according to WQI as shown in Table 1. The higher the WQI value, the more serious the pollution.

**Table 1.** Classification of water quality index.

| WQI | Water Quality Level | Classification Standard |
|---|---|---|
| ≤20 | Great (I) | Most targets were not detected, and the detected values were within the standard |
| 20–40 | Good (II) | The detected values were within the standard |
| 40–70 | Medium (III) | Most values were within the standard, and one target was close to or exceed the standard |
| 70–100 | Slightly polluted (IV) | Two detected values exceed the standard |
| >100 | Heavy polluted (V) | Most of the detected values exceeded the standard |

### 2.3.3. Analysis of Spatial Autocorrelation

Before running the geographically-weighted regression (GWR) model, it was necessary to clarify the aggregation of ditch connectivity indicators and water quality indicators. Spatial autocorrelation analysis can determine whether each ditch connectivity index and water quality index were spatially positively correlated, negatively correlated, or randomly distributed [24,25]. The spatial correlation of spatial data caused the heterogeneity of the regression relationship [26]. Therefore, spatial autocorrelation analysis was the premise to ensure the accuracy of spatially-weighted regression [27]. This study chose Moran's I index to explore the spatial distribution and agglomeration status of each index. The calculation formula is as follows:

$$I = \frac{1}{\sum_{i=1}^{n} \sum_{j=1}^{n} W_{ij}} \times \frac{\sum_{i=1}^{n} \sum_{j=1}^{n} W_{ij}(Z_i - Z)}{\sum_{i=1}^{n} (Z_i - Z)\frac{2}{n}} \qquad (11)$$

where $n$ is the number of spatial units, $Z_i$ and $Z_j$ are the attribute values of spatial units $i$ and $j$, $W_{ij}$ is the spatial weight of the spatial units. The value of $I$ is between $-1$ and 1. $I > 0$ indicates a positive spatial correlation. The closer the $I$ value is to 1, the stronger the positive spatial correlation. $I < 0$ indicates a negative spatial correlation. $I = 0$ means the randomly distributed. Moran's I test statistic is:

$$Z(I) = \frac{I - E(I)}{\sqrt{Var(I)}} \qquad (12)$$

The statistic $Z(I)$ was used to judge the statistically significance of the Moran's I index. The $E(I)$ was the theoretical mathematical expectation and $Var(I)$ was the theoretical variance. If the $Z(I)$ value of the normal statistic of Moran's I was greater than the critical

value of 1.65, which was the normal distribution function at the 0.05 level, it indicated that the observed value had a significant positive correlation in the spatial distribution, if the $Z(I)$ value of the normal statistics of Moran I's was greater than the critical value of 2.58, which was the normal distribution function at the 0.01 level, it indicated that the observed value had a strong positive correlation in the spatial distribution, which revealed a clustering distributed of the eigenvalues. The eigenvalue is the value that the attribute value of the spatial unit is different from the surrounding in space. This analysis was the premise for ensuring the validity of the GWR model.

### 2.3.4. Geographically Weighted Regression (GWR) Analysis

The spatial autocorrelation analysis indicated the agglomeration of the ditch connectivity index and the water quality index in the spatial distribution. This condition can ensure the feasibility of the GWR model, which was an improvement of the ordinary linear regression model. Its principle was to compare and analyze the data of a certain variable with other variables in the neighboring area. The calculated value of the model changed with geographic location [25], so the heterogeneity was found in different locations [28,29]. This study conducted a GWR analysis between the ditch connectivity indicators and the water quality index to explore the relationship. The model formula was as follows:

$$y_i = \beta_0(u_i, v_i) + \beta_1(u_i, v_i)I_{di} + \beta_2(u_i, v_i)\alpha_i + \beta_3(u_i, v_i)\gamma_i + \beta_4(u_i, v_i)\delta_i + \varepsilon_i \tag{13}$$

where $(u_i, v_i)$ was the geographic coordinates of the *i*-th cell; $y$ was the dependent variable of water quality index; $I_{di}$, $\alpha_i$, $\gamma_i$, $\delta_i$ were the independent variable of the ditch density ($I_d$), circularity ($\alpha$), network connectivity ($\gamma$), water conservancy project index ($\delta$) at cell $(u_i, v_i)$. $\varepsilon_i$ was a random error. $\beta_j(u_i, v_i)$ is the *j*-th regression coefficient of the *i*-th cell. The positive and negative of $\beta_j(u_i, v_i)$ indicate the positive or negative effect on $y_i$, respectively.

## 3. Results

### 3.1. The Spatial Difference of the Ditch Connectivity

The spatial differences of ditch connectivity in the study area were shown in Figure 3. The classification method of natural break was used to divide the indicators into five classes. Low-value of $I_d$ were mainly distributed in urban areas, suburb, and transition zone while high-value of $I_d$ was mainly located in rural areas. The $I_d$ increased gradually from urban to rural (Figure 3a). In Figure 3b, the value of $\alpha$ was between 0 and 1. The low value and the high value were mainly distributed in the urban area and rural area, respectively. The value of $\gamma$ increased from urban area to rural area in Figure 3c and the main reason was that the number of ditches in urban areas was smaller than in rural areas. Agricultural land, such as farming ponds and paddy fields, was the main land use type in rural areas and water resources where delivery though ditches was needed to meet the demand of agricultural lands in rural areas. The value of $\delta$ was also reduced gradually from rural to urban areas, and water conservancy projects were mostly used to enhance the circulation of water resources.

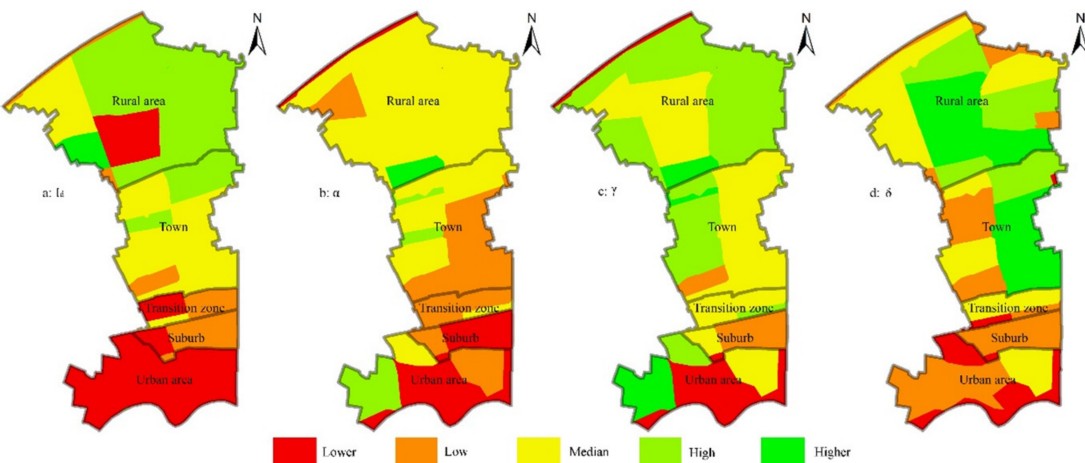

**Figure 3.** Distribution map of the connectivity index values of each ditch in the study area. (**a**) ($I_d$): ditch density; (**b**) ($\alpha$): circularity; (**c**) ($\gamma$): network connectivity; (**d**) ($\delta$): water conservancy project index.

### 3.2. Spatial Heterogeneity of Water Quality Index

The water quality index of each region in the study area is shown in Figure 4. The water quality index of the suburbs was 160.66, which was the worst water quality within category V [30]. The water quality index in rural areas was 56.55, which was the best water quality. The water quality indexes of urban areas, transition zones, and towns were similar, but the DO, $NH_4^+$-N, and TN changed greatly in each region, which affected the water quality index.

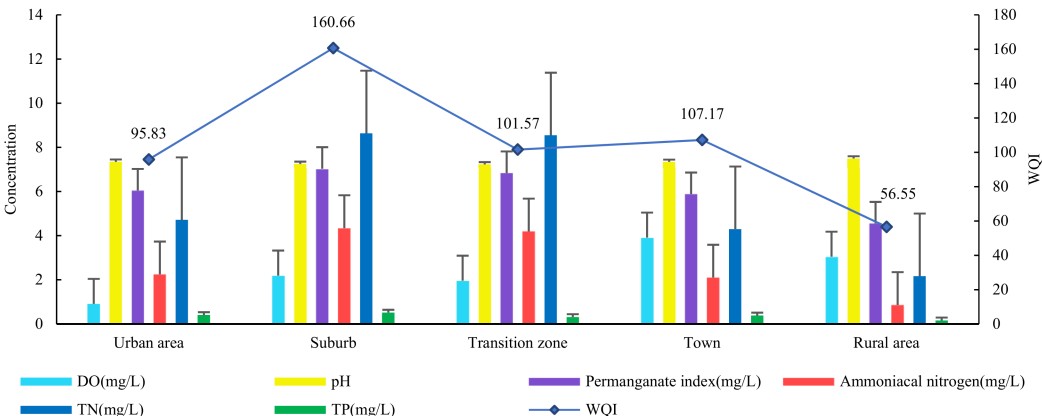

**Figure 4.** Water quality index and single index of ditches in the study area.

The spatial difference of the water quality index between urban and rural areas is shown in Figure 5. The urban area was dominated by III, IV, and V of water quality. The water in southern urban areas was well connected to the Sheyang River, which explained why water in southern urban areas was better than the northern urban area. Water quality in suburban and transition zones was class IV and V. The domestic sewage in suburban and transition zone was not completely treated and discharged to the ditches, resulting in poor water quality. The water quality in the towns was mainly classified as class III and IV. There were many water conservancy projects such as pumping stations, culvert, and sluice in the towns. The large amount of water used for agricultural land resulted in better water circulation and better water quality. The water quality in rural areas was mostly classified as class III, which was the best water quality in the study area. The large number of ditches in rural areas had significant impacts on the water quality. On the other hand, the ditches in rural areas were connected with the northern Jiangsu irrigation canal, which has water quality of II class. In general, the spatial difference of the water quality showed a

gradient increase from urban to rural areas, and water quality deteriorated in the suburbs and transition zones.

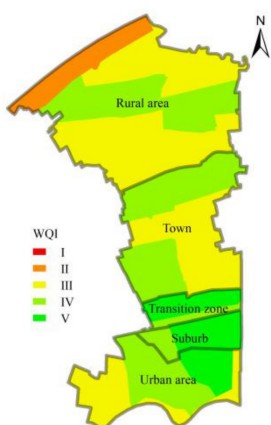

**Figure 5.** Spatial differences of the water quality index in the study area.

*3.3. Regression Model of Ditch Connectivity Index and Water Quality Index*

For the GRW model, the water quality index was used as the dependent variable, and the ditch connectivity index was used as the independent variable. The regression coefficients of each independent variable in the GWR model were listed in Table 2. The circularity of ditches had the stable negative influence on the water quality index. The network connectivity of ditches was significantly positively correlated with water quality index but had no correlation between water conservancy project index and the water quality index. Under the conditions that other influencing factors remain unchanged, the water quality index would drop by 3 with the 10 km/km$^2$ increase in ditch density. The water quality index would drop by 16.6 for every 0.1 increase in the ditch circularity. For every 0.1 increase in the connectivity of the ditches, the water quality index would increase by 12.0. When the water conservancy project index increased by 1, the water quality index would drop by 0.71.

**Table 2.** Statistical analysis of regression coefficients of GWR model.

| Independent Variable | Average | Maximum | Minimum | Upper Quartile | Lower Quartile | Median |
|---|---|---|---|---|---|---|
| $I_d$ | −0.22 | 0.21 | −0.44 | −0.38 | −0.06 | −0.31 |
| $\alpha$ | −168.66 | −147.58 | −201.69 | −179.08 | −156.43 | −166.32 |
| $\gamma$ | 118.94 | 141.86 | 95.50 | 105.65 | 129.65 | 120.25 |
| $\Delta$ | −0.63 | −0.09 | −0.91 | −0.85 | −0.43 | −0.71 |

$I_d$: ditch density; $\alpha$: circularity; $\gamma$: network connectivity; $\delta$: water conservancy project index.

The regression coefficients of the significant influencing factors ($\alpha$ and $\gamma$) in Table 2 are analyzed in spatial difference, and the results were shown in Figure 6.

The regression coefficient of $\alpha$ was negative. The higher the $\alpha$ value, the lower the water quality index, and the better the water quality. The good circumstances of the ditches represented the high nutrient exchange capacity at the ditch intersection and the complexity of the ditch network. The low values of $\alpha$ regression coefficients were mainly distributed in the urban, suburban areas, and some rural areas where closed to the irrigation canal in northern Jiangsu. This indicated that the $\alpha$ had the most significant impact on the urban and transition zones.

The regression coefficient of $\gamma$ was positive, indicating that $\gamma$ was positively correlated with the water quality index. The higher the $\gamma$, the higher the water quality index, the worse the water quality. The high network connectivity of the ditches indicated that better connectivity in a small scale did not generate better water quality, but caused poor quality

water, flowing into good water quality, and making it worse. The gradual increase in $\gamma$ value from south to north indicated that the network connectivity index had the highest impact on the water quality index in rural areas.

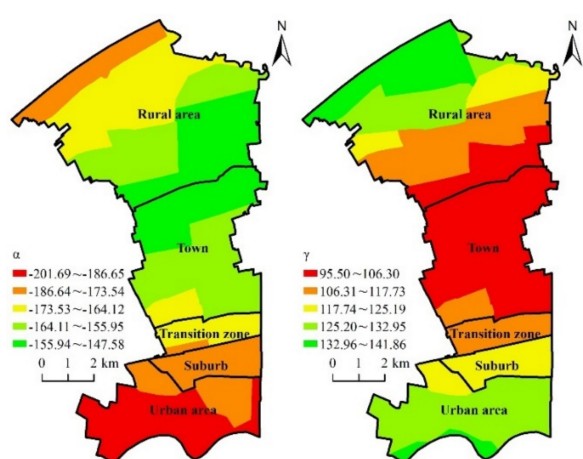

**Figure 6.** Spatial distribution of regression coefficients of ditch connectivity index in GWR model. $\alpha$: circularity; $\gamma$: network connectivity.

## 4. Discussion

### 4.1. Spatial Autocorrelation Analysis of Ditch Connectivity Indicators

In order to explore the spatial distribution characteristics of ditch connectivity, this study conducted spatial autocorrelation analysis on the ditch density index, circularity, network connectivity, and water conservancy project index in the study area. The inverse distance weight method was selected to calculate the normal statistic *z* of Moran's I, which was used to test its significance (Figure 7).

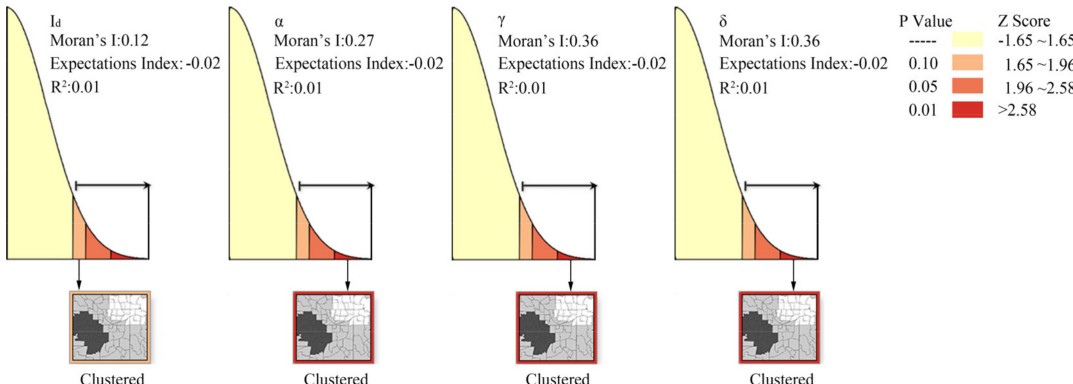

**Figure 7.** Spatial autocorrelation analysis report of each index in the study area. $I_d$: ditch density; $\alpha$: circularity; $\gamma$: network connectivity; $\delta$: water conservancy project index.

The Moran's I index of the four ditch indicators were all positive and greater than the expected index. The z value of $I_d$ was 1.71, which was in the range of 1.65~1.96. This meant that the probability of random distribution was less than 10%, and the probability of data clustering was greater than that of random distribution. The z-scores of $\alpha$, $\gamma$, and $\delta$ were 3.14, 3.95, and 3.59, respectively, which were all greater than 2.58. This demonstrated that the probability of random distribution was less than 1%. The probability of clustering was greater than that of random distribution, and the null hypothesis can be significantly rejected. The results showed that the spatial distribution of the ditch connectivity indicators had a certain clustering feature, and the spatial correlation was significantly positive.

The spatial characteristics of hydrological connectivity is mainly based on the perspective of landscape ecology on the current research [31,32]. Based on the division of spatial units in the research area, the spatial heterogeneity of hydrological connectivity was calculated and analyzed [33]. Because of similar land use conditions and human activities in the study regions, the research showed that hydrological connectivity indicators have spatial aggregation on the space [34]. For example, the rural area is dominated by agricultural land, and the ditches in this area are used for farmland irrigation. Urban areas are dominated by construction land, and the ditches in this area are used for flood protection.

*4.2. Spatial Autocorrelation Analysis of Water Quality Index*

The inverse distance weighting method was used to simulate the spatial autocorrelation of the water quality index. Normal statistic *z* of Moran's I was obtained to test the autocorrelative significance. The results showed that the Moran's I index of water quality was 0.66, which was greater than the expected index of −0.02. The spatial autocorrelation z score of the water quality index was 7.14, which was greater than 2.58. The probability of random distribution was less than 1%. The water quality index in the study area had a certain clustering characteristic in the spatial distribution, and the spatial correlation was significantly positive (Table 3).

**Table 3.** Spatial autocorrelation analysis of the water quality index in the study area.

| | Moran's I | Expectation Index | $R^2$ | Z Score | *p* Value | Result |
|---|---|---|---|---|---|---|
| Water quality index | 0.66 | −0.02 | 0.01 | 7.14 | 0.00 | Clustering |

The water quality evaluation methods mainly include single factor evaluation, comprehensive pollution index evaluation, Canadian water quality index method, Nemerow Synthetical Pollution Index, etc. [35–37] The single-factor evaluation used the most severely polluted index of water quality as the evaluation result, and the evaluation result would be worse than the real result [38]. Compared with the single-factor evaluation, the comprehensive pollution index method used more comprehensive water quality indicators to assess water quality. The Canadian Water Quality Index is generally used to analyze the water quality of drinking water sources, which was obtained through the analysis of a long series of data [39,40]. The Nemeiro Pollution Index method used the actual measured concentration and standard concentration of selected indicators, and relates with the standard value of Class III water as the benchmark [41]. The time and space analysis of water quality mostly used cluster analysis and spatial autocorrelation analysis [42,43]. Studies have shown that the spatial similarity and cluster of water quality was consistent, which was highly related with land use density and human activities. For example, pollution by domestic sewage, pesticides, and fertilizers in rural areas. Water pollutions were caused by similar land use types and similar human activities, resulting in spatial agglomeration of the water quality.

*4.3. GWR Goodness of Fit and Residual Distribution*

The ditch connectivity indicators, namely ditch density, circularity, network connectivity, and water conservancy project index, were used as independent variables, and the water quality index was used as the dependent variable. A geographically-weighted regression model (GWR) was constructed by ArcGIS 10.2. The estimated results of the model parameters were shown in Table 4. The model explains the spatial difference of the ditch connectivity index, with an explanation degree of 63.74% of water quality changes which indicated that the water quality index can be explained by the GWR model.

**Table 4.** Parameter estimation and test results of GWR model.

| Variable | Residual Squares | Effective Number | Sigma | AICc | $R^2$ | $R^2$ Adjusted |
|---|---|---|---|---|---|---|
| Parameter | 4961.68 | 10.19 | 11.77 | 370.92 | 0.63 | 0.53 |

The residuals in the GWR model were tested for spatial autocorrelation, and the results are shown in Figure 8, where Moran's I was 0.14, Z was 1.73 and *p* was 0.10. This revealed that the residuals of the GWR model were clustered and the simulation effect of the model was good. Figure 9 was the spatial distribution of the standardized residuals of the GWR model, where the standardized residuals of the 46 units (100%) in the study area were all within [−2.5–2.5]. The distribution showed that the ditch connectivity index in the whole study area had an impact on the water quality index, and the correlation between ditch connectivity index and water quality index was significant.

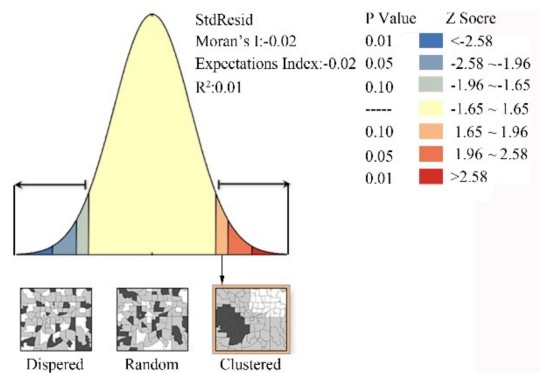

**Figure 8.** Standardized residual spatial autocorrelation report.

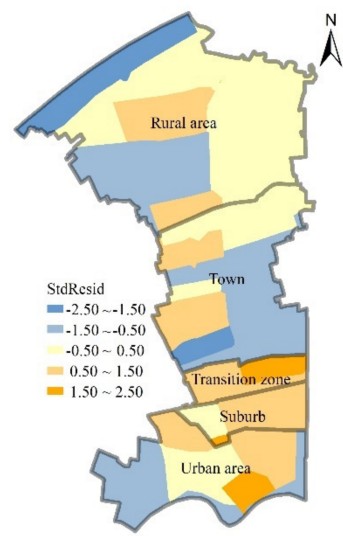

**Figure 9.** Spatial distribution of standardized residuals of GWR model.

### 4.4. The Spatial Management Patterns of Ditches to Improve Water Quality

According to the research results, this research proposed ditch optimization and management strategies for different regions:

The water was medium (III) polluted ($40 < WQI \leq 70$) in the urban area near the Sheyang River. The pollution mainly came from the discharge of domestic sewage into the Sheyang River in the urban area. In other urban areas, water quality was at slightly polluted (IV) and heavy polluted (V) levels. Water pollution in the urban areas was mainly

due to elevated $NH_4^+$-N, TN, and TP. The number of ditch nodes can be increased to extend the water retention time and improve the material exchange capacity. For every 10 ditches, increase at the intersections of the ditches in the urban, circularity ($\alpha$), and network connectivity ($\gamma$) will increase 0.22 and 0.14, respectively. The WQI will decrease by 19.53 and the water quality will change from level V to level IV, level IV to level III.

Studies have shown that as the length of the ditches increases, the flow time of sewage in the ditches is prolonged, and the total organic matter in the ditches degraded by microorganisms will increase. At present, the methods of improving water quality in urban areas included river consolidation, dredging, sewage discharge treatment, clearing plan, increasing the connection and length between ditches, and using microorganisms to treat sewage.

The WQI in the suburbs was on level IV and V. The pollution was caused by overuse of fertilizer and domestic sewage that was not completely treated. In the suburbs, the land type was complicated with a mix of residential and agricultural land. The water quality should be improved by controlling sewage discharge and increasing the number of drainage ditches in residential areas. Every 10 ditches added in the suburbs will lead to $\alpha$ and $\gamma$ increased by 0.07 and 0.05, and WQI will decrease by 19.53. When 50 ditches are added, WQI will decrease by 33.10, and the water quality will change from level IV and V to level III.

In the transition zone, all the agricultural lands, mainly planted to fruit trees and vegetables, led to pollution caused by excessive fertilization. Based on controlling fertilizer usage, the number of ditches and water conservancy projects should be increased to ensure the connectivity of the ditches and enhance the connection between the ditches and the river to improve the self-purification capacity of the ditches. For every 10 added ditches in the transition zone, the WQI decreases by 7.83. For every additional 50 ditches, WQI will decrease by 39.15. The water quality will change from level IV and V to level III.

The water quality of the towns was mainly affected by the irregular discharges of domestic sewage produced by the residents and the use of fertilizers. The number of ditches should be increased to ensure the discharge of sewage and enhance the connection between agricultural land areas and the river. However, compared to other regions, improving water quality by increasing the number of ditches in the towns may not achieve good results.

Water pollution in rural areas was mainly caused by excess $NH_4^+$-N and TN from over fertilization. The WQI was Class II and III in rural areas and the water quality was in a good state. Due to the needs of irrigation and drainage in rural areas, the system of ditches was quite developed. The connection between the ditches in the region was better than for other regions, which was the main reason for the better water quality. By increasing the number of ditches to improve water quality, the effect will not be great. For every 10 ditches added, the WQI index decreases by 0.18, but for every additional 1000 ditches, the WQI index should decrease by only 17.75. So, in rural areas, the effective way to improve the water quality may be to reduce the use of fertilizers rather than through adding ditches (Figure 10).

In addition, the measures of improving water pollution in rural areas also included farmland non-point source pollution control, livestock and poultry breeding pollution control, surface runoff sewage purification and utilization methods. Studies have shown that the treatment of livestock and poultry farming pollution, and the control of farmland non-point source pollution, in the small rural watersheds have achieved obvious effects for improving water quality [44,45].

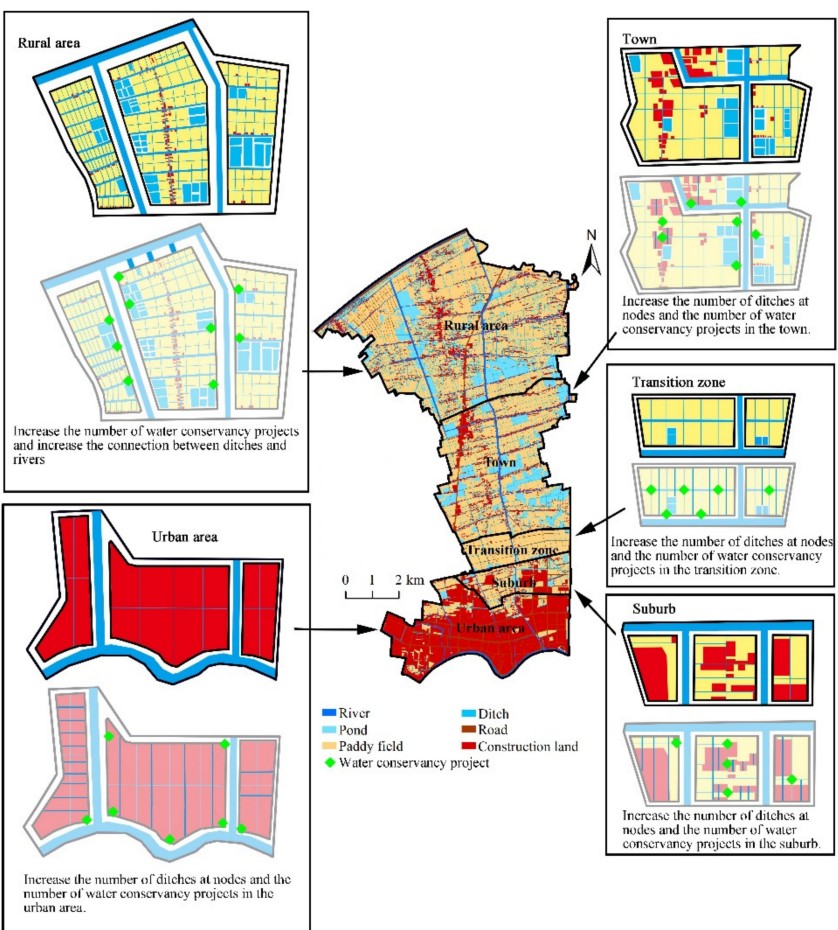

**Figure 10.** Schematic diagram of optimization management strategies for ditches in different regions.

## 5. Conclusions

In order to explore the spatial relationships between the ditch connectivity index and water quality, this study used spatial autocorrelation analysis and a geographically-weighted regression model to analyze their relationships. Optimization plans and management decisions of ditch network structure were proposed to improve water quality. The following conclusions were obtained:

The low-value of water quality index was distributed in rural areas and towns. The lowest water quality index in rural areas had the best water quality. The suburbs had the highest water quality index and the worst water quality, followed by transition zones.

The results of GWR showed that the model had a good fitting effect, with 100% units passing the residual test analysis of the model. The ditch connectivity index had a significant impact on the water quality index, and the regression coefficients of the GWR model explained that the circularity and network connectivity of the ditches had the greatest impact on water quality in urban areas and towns.

From the urban to rural areas, there were differences in ditch connectivity optimization schemes for different regions. In the urban and transitional areas, increasing the number of ditches to improve water quality was most effective. In rural areas, due to the existing great connectivity of the ditch system, increasing the number of ditches was not a good way to improve water quality.

**Author Contributions:** Conceptualization, C.Q.; methodology, C.Q., Y.Z. and H.L. software, C.Q.; validation, C.W., W.H. and Y.W.; formal analysis, C.Q. and J.X.; investigation, C.Q. and Y.L.; resources, A.L.W.; data curation, C.Q.; writing—original draft preparation, C.Q.; writing—review and editing, Y.L.; visualization, C.W.; supervision, S.Z., Y.L.; project administration, Y.L.; funding acquisition, Y.L. All authors have read and agreed to the published version of the manuscript.

**Funding:** This research was funded by the National Natural Science Foundation of China (Grant 41871188, 31570459), Key Research and Development Program of Jiangsu Province (BE2018681), The Special Fund of Natural Resources Development (Marine Scientific and Technological Renovation) (JSZRHYKJ202003).

**Institutional Review Board Statement:** Not applicable.

**Informed Consent Statement:** Not applicable.

**Data Availability Statement:** The data used to support the findings of this study are available from the corresponding author upon request.

**Acknowledgments:** We convey our gratitude for the research grant support that was kindly provided by the National Natural Science Foundation of China (Grant 41871188, 31570459), Key Research and Development Program of Jiangsu Province (BE2018681), The Special Fund of Natural Resources Development (Marine Scientific and Technological Renovation) (JSZRHYKJ202003). The data used to support the findings of this study are available from the corresponding author upon request.

**Conflicts of Interest:** The authors declare no conflict of interest.

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
