# Peer review of "Spatial Effects of Urban-Rural Ditch Connectivity Gradient Changes on Water Quality to Support Ditch Optimization and Management"

_sustainability, doi:10.3390/su13158329_

Round 1
Reviewer 1 Report
-
Author Response
Response to Reviewer 1 Comments
Dear Reviewer 1,
Thank you for reviewing this article. Based on the opinions of experts, we have made the following changes to the paper to enhance the readability of the article.
1.We have strengthened the discussion part of the article and discussed the scientific nature of the article more effectively.
Paragraph 4.1:
The spatial characteristics of hydrological connectivity is mainly based on the perspective of landscape ecology on the current research [31,32]. Based on the division of spatial units in the research area, the spatial heterogeneity of hydrological connectivity was calculated and analyzed [33]. Because of similar land use conditions and human activities in the study regions, the research showed that hydrological connectivity indicators have spatial aggregation on the space [34]. For example, the rural area is dominated by agricultural land, and the ditches in this area are used for farmland irrigation. Urban areas is dominated by construction land, and the ditches in this area are used for flood protection.
- Shen, J.Z., Jun.; Shang, Lizhao. Analysis of the impact of landscape pattern change on the river network connectivity based on the spatial auto-regressive model Journal of East China Normal University (Natural Science) 2015, 124-135.
- Jin, Y. The Study of River Network Structural Changes and Space Partition in South of Yangtze River: A Case study of Qingpu, Shanghai. . East China Normal University 2013.
- Persic, V.; Horvatic, J. Spatial Distribution of Nutrient Limitation in the Danube River Floodplain in Relation to Hydrological Connectivity. Wetlands 2011, 31, 933-944, doi:10.1007/s13157-011-0208-1.
- Zhao, D. Landslide disaster ecological risk of small watershed Honghe Hani Rice Terraces World Heritage Site by means of susceptibility and connectivity. Yunnan Normal University 2020.
Paragraph 4.2:
The water quality evaluation methods mainly include single factor evaluation, comprehensive pollution index evaluation, Canadian water quality index method, Nemerow Synthetical Pollution Index, etc [35-37] . The single-factor evaluation used the most severely polluted index of water quality as the evaluation result, and the evaluation result would be worse than the real result [38]. Compared with the single-factor evaluation, the comprehensive pollution index method used more comprehensive water quality indicators to assess water quality. The Canadian Water Quality Index is generally used to analyze the water quality of drinking water sources, which was obtained through the analysis of a long series of data [39,40]. The Nemeiro Pollution Index method used the actual measured concentration and standard concentration of selected indicators, and relates with the standard value of Class III water as the benchmark [41]. The time and space analysis of water quality mostly used cluster analysis and spatial autocorrelation analysis [42,43]. Studies have shown that the spatial similarity and cluster of water quality was consistent, which was highly related with land use density and human activities. For example, pollution by domestic sewage, pesticides and fertilizers in rural areas. Water pollutions were caused by similar land use types and similar human activities, resulting in spatial agglomeration of the water quality.
- Ha, H.J.; Hee, P.S.; Min, S.C. A Study on an Integrated Water Quantity and Water Quality Evaluation Method for the Implementation of Integrated Water Resource Management Policies in the Republic of Korea. Water 2020, 12.
- Wang, W. The Application of Different Water Quality Evaluation Methods in Water Quality Assessment of Lancang River Source Region. Journal of Low Carbon Economy 2015, 4.
- Bhat, S.U.; Pandit, A.K. Water quality assessment and monitoring of Kashmir Himalayan freshwater springs-A case study. Aquatic Ecosystem Health & Management 2020, 23, 274-287, doi:10.1080/14634988.2020.1816771.
- Zheng, K.Z., Lei.; Xue, Chenliang. Application research of single factor index method in water quality evaluation. Groundwater 2018, 40, 79-80.
- Guo, J.W., Chongming.; Huang, Daizhong.; Li, Liqiang.; Lian, Hua. Pollution characterization and water quality assessment of Dongting Lake. Environmental Chemistry 2019, 38, 152-160.
- Mohammed, F.M.A.; Saadi, R.J.M.A.; Fawzy, A.M.A.; Ali, S.H.M.; Mutasher, A.K.A.; Hommadi, A.H. The Analysis of Water Quality Using Canadian Water Quality Index: Green Belt Project/Kerbala-Iraq. International Journal of Design & Nature and Ecodynamics 2021, 16.
- Ning, Y.; Yin, F. Water quality evaluation based on improved Nemerow pollution index method and grey clustering method. Journal of Central China Normal University. Natural Sciences Edition 2020, 54, 149-155.
- Gagandeep, K.; Richa, K.; Sunil, D.; Deepak, P.; V, T.V. Impact assessment on water quality in the polluted stretch using a cluster analysis during pre- and COVID-19 lockdown of Tawi river basin, Jammu, North India: an environment resiliency. Energy, ecology & environment 2021.
- Ding, Y.; Zhao, J.-Y.; Zhang, J.; Fu, Y.-C.; Peng, W.-Q.; Chen, Q.-C.; Li, Y.-Y. Spatial Differences in Water Quality and Spatial Autocorrelation Analysis of Eutrophication in Songhua Lake. Huan jing ke xue= Huanjing kexue 2021, 42, 2232-2239, doi:10.13227/j.hjkx.202009244.
Paragraph 4.4:
Studies have shown that as the length of the ditches increases, the flow time of sewage in the ditches is prolonged, and the total organic matter in the ditches degraded by microorganisms will increase. At present, the methods of improving water quality in urban areas included river consolidation, dredging, sewage discharge treatment, clearing plan, increasing the connection and length between ditches and using microorganisms to treat sewage.
In addition, the measures of improving water pollution in rural areas also included farmland non-point source pollution control, livestock and poultry breeding pollution control, surface runoff sewage purification and utilization methods. Studies have shown that the treatment of livestock and poultry farming pollution and the control of farmland non-point source pollution in the small rural watersheds have achieved obvious effects for improving water quality [44,45].
- Liu, S. Situation and Control Effect of Agricultural Non-point Source Pollution in Typical Small Watershed of Fengle River. Anhui Agricultural University, 2020.
- Yang, Y. Quality Assessment and Improvement Countermeasures of Countryside Drinking Water Source In Taian City. Shandong Agricultural University, 2008.
- In the revised manuscript, We have added titles and logos to Figures 3 and 10 for easy reading
3.In the revised manuscript, we have revised the citations and references throughout the article.
- In the revised manuscript, we have revised the English spelling of the whole article.
- Finally, We have re-explained the roles and contributions of the authors in the system.

Reviewer 2 Report
Sustainability
Manuscript Number: sustainability-1280831
Title: Spatial effects of urban-rural ditch connectivity gradient changes on water quality to support ditch optimization and management
Reviewer' comments:
Overall statement of the article
This paper considered ditch connectivity and water quality indicators comprehensively using spatial autocorrelation and geographically weighted regression (GWR) models to analyze the impact of ditch connectivity on water quality from urban to rural gradients.
The paper contains 5 keywords: urban-rural gradient; ditch connectivity; spatial autocorrelation; GWR model; water quality.
This work is divided into 5 distinct parts: (i) Abstract, (ii) Introduction, (iii) Materials and Methods, (iv) Results, (v) Discussion, (vi) Conclusion.
Materials and Methods part is subdivided into 3 main sub-parts : a. Study area, b. Datasets, c. Methodology. The Results’ section includes three sub-parts: a. The spatial difference of the ditch connectivity, b. Spatial heterogeneity of water quality index, c. Regression Model of Ditch Connectivity Index and Water Quality Index. The discussions are presented in a. Spatial autocorrelation analysis of ditch connectivity indicators, b. Spatial autocorrelation analysis of water quality index, c. GWR goodness of fit and residual distribution, d. The spatial management patterns of ditches to improve water quality.
The paper has 4 tables and 10 figures. There are 32 references, which are from 1994 to 2021. The majority of references are from recent dates, and are internationally evaluated and published in peer-reviewed journals with important impact factors.
The following conclusions were obtained by the authors for this study:
- The low-value of water quality index was distributed in rural areas and towns. And the lowest water quality index in rural areas had the best water quality. The suburbs had the highest water quality index and the worst water quality, followed by suburbs.
- The results of GWR showed that the model had a good fitting effect, with 100% units passing the residual test analysis of the model. The ditch connectivity index had a significant impact on the water quality index And the regression coefficients of the GWR model explained that the circularity and the network connectivity of the ditches had the greatest impact on water quality in urban areas and towns.
- From the urban to rural areas, there were differences in ditch connectivity optimization schemes for different regions. In the urban and transitional areas, increasing the number of ditches to improve water quality was most effective. In rural areas, due to the existing great connectivity of the ditch system,
Overall strengths of the article
The title is clear and the authors clearly state the aims of the paper sustainability-1280831. The topic of the works is interesting, and the article reflects a present state of knowledge with a literature sufficiently current and internationally evaluated. The paper has very relevant results. It is an informative paper. The authors provide information on the subject, and do not discuss sufficiently information reported in the scientific literature.
Methodology
The methodology is very well developed.The variables have been well defined. The methods are valid and reliable.
Overall statement
In order to make the article more convincing, the reviewer asks the authors to revise the article by strengthening it in the discussions section.
In fact, in none of the 3 subsections of the discussion did the authors compare their results with those available in the literature. No bibliographic reference is mentioned in the discussions.
Authors are requested to reference Figure 10 and add its title to the manuscript.
The reviewer asks the authors to re-consult the review author's guide and to check the citations in the manuscript as well as the list of references.

Author Response
Response to Reviewer 2 Comments
Dear Reviewer 2,
Thank you for your comments on the manuscript. Your constructive comments helped to clarify the concepts of the paper. According to your comments, we made modifications to the manuscript. Our detailed responses to your comments are below.
Point 1:
In order to make the article more convincing, the reviewer asks the authors to revise the article by strengthening it in the discussions section.
In fact, in none of the 3 subsections of the discussion did the authors compare their results with those available in the literature. No bibliographic reference is mentioned in the discussions.
Response 1: Thank you for your suggestion.A discussion section has been added to the revised manuscript(in red):
Paragraph 4.1:
The spatial characteristics of hydrological connectivity is mainly based on the perspective of landscape ecology on the current research [31,32]. Based on the division of spatial units in the research area, the spatial heterogeneity of hydrological connectivity was calculated and analyzed [33]. Because of similar land use conditions and human activities in the study regions, the research showed that hydrological connectivity indicators have spatial aggregation on the space [34]. For example, the rural area is dominated by agricultural land, and the ditches in this area are used for farmland irrigation. Urban areas is dominated by construction land, and the ditches in this area are used for flood protection.
Paragraph 4.2:
The water quality evaluation methods mainly include single factor evaluation, comprehensive pollution index evaluation, Canadian water quality index method, Nemerow Synthetical Pollution Index, etc [35-37] . The single-factor evaluation used the most severely polluted index of water quality as the evaluation result, and the evaluation result would be worse than the real result [38]. Compared with the single-factor evaluation, the comprehensive pollution index method used more comprehensive water quality indicators to assess water quality. The Canadian Water Quality Index is generally used to analyze the water quality of drinking water sources, which was obtained through the analysis of a long series of data [39,40]. The Nemeiro Pollution Index method used the actual measured concentration and standard concentration of selected indicators, and relates with the standard value of Class III water as the benchmark [41]. The time and space analysis of water quality mostly used cluster analysis and spatial autocorrelation analysis [42,43]. Studies have shown that the spatial similarity and cluster of water quality was consistent, which was highly related with land use density and human activities. For example, pollution by domestic sewage, pesticides and fertilizers in rural areas. Water pollutions were caused by similar land use types and similar human activities, resulting in spatial agglomeration of the water quality.
Paragraph 4.4:
Studies have shown that as the length of the ditches increases, the flow time of sewage in the ditches is prolonged, and the total organic matter in the ditches degraded by microorganisms will increase. At present, the methods of improving water quality in urban areas included river consolidation, dredging, sewage discharge treatment, clearing plan, increasing the connection and length between ditches and using microorganisms to treat sewage.
In addition, the measures of improving water pollution in rural areas also included farmland non-point source pollution control, livestock and poultry breeding pollution control, surface runoff sewage purification and utilization methods. Studies have shown that the treatment of livestock and poultry farming pollution and the control of farmland non-point source pollution in the small rural watersheds have achieved obvious effects for improving water quality [44,45].
Point 2:
Authors are requested to reference Figure 10 and add its title to the manuscript.
Response 2: Thank you for your suggestion. Figure 10 has been captioned in the revised draft.(in red)
Figure 10. Schematic diagram of optimization management strategies for ditches in different regions
Point 3:
The reviewer asks the authors to re-consult the review author's guide and to check the citations in the manuscript as well as the list of references.
Response 3: Thank you for your suggestion. Revise the manuscript to revise the citations and reference lists in the text.

Reviewer 3 Report
This contribution is aimed to analyze the differences in the characteristics of ditch connectivity under the urban-rural gradient, and to explore the spatial relationship between ditch connectivity and water quality through spatial autocorrelation analysis and geographically-weighted regression models. The presented research proposed ditch optimization and management strategies for the different considered regions.
In general, the paper is well-written and it is significant for the journal. I have only a few remarks that authors should address before acceptance
- English is good with only minor typing mistakes. Nevertheless, a thorough English revision is recommended.
- Line 93: What it understood by "urban and rural gradient changes"?
- Line 110: I think that “suburbs” should be replaced by “transition zones”
- In general, it is recommended to improve the quality of all the formulae.
- Line 233: In this context, what the eigenvalues are? Please, clarify
- Line 260: Fig 3c is not identified. Please. label the subfigures in Fig. 3.
- Line 263: I think that the word “were” should be “where”.
- Line 271: replace “was” by “is”. In general the presentation of results in figures should be done in present, not in past. This fact happens in nearly all the figures description.
- Line 279: replace “was” by “is”.
- Line 313: replace “were” by “are”.
- Line 345: “was indicated” should be replaced by “meant” or “indicated”.
- Line 374: replace “were” by “are”.
- Line 438: Sentence “The suburbs had the highest water quality index and the worst water quality, followed by suburbs” should be checked as “suburbs” appears twice.
Finally, authors taxonomy section is missing. As there are 10 authors of the paper, their roles and contributions should be clearly specified.
Therefore, I recommend publishing the paper once that the previous comments have been addressed.
Author Response
Response to Reviewer 3 Comments
Dear Reviewer 3,
Thank you for your comments on the manuscript. Your constructive comments helped to clarify the concepts of the paper. According to your comments, we made modifications to the manuscript. Our detailed responses to your comments are below.
Point 1: English is good with only minor typing mistakes. Nevertheless, a thorough English revision is recommended.
Response 1: Thank you for your suggestion. In the revised manuscript, we have revised the English spelling of the whole article.
Point 2: Line 93: What it understood by "urban and rural gradient changes"?
Response 2: The urban and rural gradient changes are the different areas divided due to the different land use conditions in the study area, which affects the distribution of ditches in the study area. In the article 2.2.2. Urban-rural gradient division has a detailed explanation on the division of urban-rural gradient.
Point 3: Line 110: I think that “suburbs” should be replaced by “transition zones”
Response 3: Thanks for the suggestion. In the revised draft, we have made changes.
Point 4: In general, it is recommended to improve the quality of all the formulae.
Response 4: Thanks for the suggestion. In the revised manuscript, we revised the formula to improve the clarity for readers to read.
Point 5: Line 233: In this context, what the eigenvalues are? Please, clarify
Response 5: Thanks for your suggestion. The eigenvalue is the value that the attribute value of the spatial unit is different from the surrounding in space. Explained in lines 257~258 in the article (in red).
Point 6: Line 260: Fig 3c is not identified. Please. label the subfigures in Fig. 3.ï¼›
Response 6: Thanks for your suggestion. In the revised manuscript, we have added a, b, c, and d to Figure 3 for the convenience of readers.
Point 7~12:Line 263: I think that the word “were” should be “where”.
Line 271: replace “was” by “is”. In general the presentation of results in figures should be done in present, not in past. This fact happens in nearly all the figures description.
Line 279: replace “was” by “is”.
Line 313: replace “were” by “are”.
Line 345: “was indicated” should be replaced by “meant” or “indicated”.
Line 374: replace “were” by “are”.
Response 7~12: Thank you for the suggestion. In the revised draft, we have made changes.
Point 13: Line 438: Sentence “The suburbs had the highest water quality index and the worst water quality, followed by suburbs” should be checked as “suburbs” appears twice.
Response 13: Thank you for your kind suggestion. In the revised draft, we have changed the suburbs into transitional areas.
Point 14: Finally, authors taxonomy section is missing. As there are 10 authors of the paper, their roles and contributions should be clearly specified.
Response 14: Thank you for your suggestions. We have re-explained the roles and contributions of the authors in the submission system.

Round 2
Reviewer 2 Report
Sustainability Manuscript Number: sustainability-1280831-v2 Title: Spatial effects of urban-rural ditch connectivity gradient changes on water quality to support ditch optimization and management Reviewer' comments: Overall statement of the article This paper considered ditch connectivity and water quality indicators comprehensively using spatial autocorrelation and geographically weighted regression (GWR) models to analyze the impact of ditch connectivity on water quality from urban to rural gradients. The following conclusions were obtained by the authors for this study: 1. The low-value of water quality index was distributed in rural areas and towns. And the lowest water quality index in rural areas had the best water quality. The suburbs had the highest water quality index and the worst water quality, followed by suburbs. 2. The results of GWR showed that the model had a good fitting effect, with 100% units passing the residual test analysis of the model. The ditch connectivity index had a significant impact on the water quality index And the regression coefficients of the GWR model explained that the circularity and the network connectivity of the ditches had the greatest impact on water quality in urban areas and towns. 3. From the urban to rural areas, there were differences in ditch connectivity optimization schemes for different regions. In the urban and transitional areas, increasing the number of ditches to improve water quality was most effective. In rural areas, due to the existing great connectivity of the ditch system, Overall strengths of the article The title is clear and the authors clearly state the aims of the paper sustainability-1280831-v2. The topic of the works is interesting, and the article reflects a present state of knowledge with a literature sufficiently current and internationally evaluated. The paper has very relevant results. It is an informative paper. The authors provide information on the subject, and discuss information reported in the scientific literature. Overall statement The authors replied to the various comments made by the reviewers. They justified their answers and scrupulously worked on each of the comments. The citations in the manuscript as well as the list of references have been checked by the authors. The authors have added the Figure 10’s title into the manuscript. The authors compared their results with those available in the literature, and supported their analyzes with quotes mentioned in the discussions.

Author Response
Dear Reviewer 2,
Thank you for your suggestions on the article. Although you did not put forward any new comments in your reply, I still checked the format carefully according to the journal's requirements.
